# Mitochondrial genome sequencing of marine leukaemias reveals cancer contagion between clam species in the Seas of Southern Europe

Daniel Garcia-Souto[1,2,3]*[†], Alicia L Bruzos[1,2]†, Seila Diaz[1]†, Sara Rocha[4], Ana Pequeño-Valtierra[1], Camila F Roman-Lewis[5], Juana Alonso[5,6], Rosana Rodriguez[7], Damian Costas[7], Jorge Rodriguez-Castro[1], Antonio Villanueva[7], Luis Silva[8], Jose Maria Valencia[9,10], Giovanni Annona[11], Andrea Tarallo[11], Fernando Ricardo[12], Ana Bratoš Cetinić[13], David Posada[5,6,14], Juan Jose Pasantes[14,15], Jose MC Tubio[1,2]*

[1]Genomes and Disease, Centre for Research in Molecular Medicine and Chronic Diseases (CIMUS), Universidade de Santiago de Compostela, Santiago de Compostela, Spain; [2]Department of Zoology, Genetics and Physical Anthropology, Universidade de Santiago de Compostela, Santiago de Compostela, Spain; [3]Cancer Ageing and Somatic Mutation Programme, Wellcome Sanger Institute, Cambridge, United Kingdom; [4]Phylogenomics Lab, Universidade de Vigo, Vigo, Spain; [5]CINBIO, Universidade de Vigo, Vigo, Spain; [6]Galicia Sur Health Research Institute (IIS Galicia Sur), SERGAS-UVIGO, Vigo, Spain; [7]Centro de Investigación Mariña, Universidade de Vigo, ECIMAT, Vigo, Spain; [8]Instituto Español de Oceanografía (IEO), Centro Oceanográfico de Cádiz, Cádiz, Spain; [9]Laboratori d'Investigacions Marines i Aqüicultura, (LIMIA) - Govern de les Illes Balears, Port d'Andratx, Balearic Islands, Spain; [10]Instituto de Investigaciones Agroambientales y de Economía del Agua (INAGEA) (INIA-CAIB-UIB), Palma de Mallorca, Balearic Islands, Spain; [11]Stazione Zoologica Anton Dohrn, Napoli, Italy; [12]ECOMARE, Centre for Environmental and Marine Studies (CESAM), Department of Biology, University of Aveiro, Santiago University Campus, Aveiro, Portugal; [13]Department of Aquaculture, University of Dubrovnik, Dubrovnik, Croatia; [14]Department of Biochemistry, Genetics and Immunology, Universidade de Vigo, Vigo, Spain; [15]Centro de Investigación Mariña, Universidade de Vigo, Vigo, Spain

*For correspondence:
daniel.garcia.souto@usc.es (DG-S);
jose.mc.tubio@usc.es (JMCT)

†These authors contributed equally to this work

Competing interest: The authors declare that no competing interests exist.

**Abstract** Clonally transmissible cancers are tumour lineages that are transmitted between individuals via the transfer of living cancer cells. In marine bivalves, leukaemia-like transmissible cancers, called hemic neoplasia (HN), have demonstrated the ability to infect individuals from different species. We performed whole-genome sequencing in eight warty venus clams that were diagnosed with HN, from two sampling points located more than 1000 nautical miles away in the Atlantic Ocean and the Mediterranean Sea Coasts of Spain. Mitochondrial genome sequencing analysis from neoplastic animals revealed the coexistence of haplotypes from two different clam species. Phylogenies estimated from mitochondrial and nuclear markers confirmed this leukaemia originated in striped venus clams and later transmitted to clams of the species warty venus, in which it survives as a contagious cancer. The analysis of mitochondrial and nuclear gene sequences supports all studied tumours belong to a single neoplastic lineage that spreads in the Seas of Southern Europe.

## Editor's evaluation

This paper describes a previously unknown lineage of transmissible cancer in a clam, in which the cancer arose from a different, but related, species. The data are clear and overall, the conclusions well-supported, and this finding increases our understanding of transmissible cancers in nature and will be of broad interest.

## Introduction

Cancers are clonal cell lineages that arise due to somatic changes that promote cell proliferation and survival (*Stratton et al., 2009*). Although natural selection operating on cancers favours the outgrowth of malignant clones with replicative immortality, the continued survival of a cancer is generally restricted by the lifespan of its host. However, clonally transmissible cancers – from now on, transmissible cancers – are somatic cell lineages that are transmitted between individuals via the transfer of living cancer cells, meaning that they can survive beyond the death of their hosts (*Murchison, 2008*). Naturally occurring transmissible cancers have been identified in dogs (*Murgia et al., 2006*; *Murchison et al., 2014*; *Baez-Ortega et al., 2019*), Tasmanian devils (*Murchison et al., 2012*; *Pye et al., 2016*) and, more recently, in marine bivalves (*Metzger et al., 2015*; *Metzger et al., 2016*; *Yonemitsu et al., 2019*).

Hemic neoplasia (HN), also called disseminated neoplasia, is a type of leukaemia cancer found in multiple species of bivalves, including oysters, mussels, cockles, and clams (*Carballal et al., 2015*). Although these leukaemias represent different diseases across bivalve species, they have been classically grouped under the same term because neoplastic cells share morphological features (*Carballal et al., 2015*). Some HNs have been proven to have a clonal transmissible behaviour (*Metzger et al., 2015*), in which neoplastic cells, most likely haemocytes (i.e. the cells that populate the haemolymph and play a role in the immune response), are likely to be transmitted through marine water. In late stages of the disease, leukaemic cells invade the surrounding tissues and, generally, animals die because of the infection (*Carballal et al., 2015*), although remissions have also been described (*Burioli et al., 2019*). Despite the observation that leukaemic cells are typically transmitted between individuals from the same species, on occasion they can infect and propagate across populations from a second, different bivalve species (*Metzger et al., 2016*; *Yonemitsu et al., 2019*). Hence, these cancers represent a potential threat for the ecology of the marine environment, which argues for the necessity of their identification and characterization for their monitoring and prevention.

Here, we use multiplatform next-generation genome sequencing technologies, including Illumina short reads and Oxford Nanopore long reads, together with cytogenetics, electron microscopy, and cytohistological approaches to identify, characterize, and decipher the evolutionary origin of a new marine leukaemia that is transmitted between two different clam species that inhabit the Seas of Southern Europe, namely warty venus (*Venus verrucosa*) and striped venus (*Chamelea gallina*) (*Video 1*).

## Results and discussion

We investigated the prevalence of HN in the warty venus clam (*V. verrucosa*), a saltwater bivalve found in the Atlantic Coast of Europe and the Mediterranean Sea. We collected 345 clam specimens from six sampling regions in the Atlantic and the Mediterranean coasts of Europe across five different countries, including Spain, Portugal, France, Ireland, and Croatia (*Figure 1a*; *Supplementary file 1*). Cytohistological examination identified HN-like tumours in eight specimens from two sampling points in Spain (*Figure 1b–e*; *Figure 1—figure supplement 1*). Three HN-positive specimens (ERVV17-2995, ERVV17-2997, and ERVV17-3193) were collected in Galicia, northwest of the Iberian Peninsula in the Atlantic Ocean, and another five specimens (EMVV18-373, EMVV18-376, EMVV18-391, EMVV18-395, and EMVV18-400) were collected in the Balearic Islands, bathed by the Mediterranean Sea (*Figure 1a*; *Supplementary file 1*). Four of these specimens (ERVV17-2995, ERVV17-3193, EMVV18-391, and EMVV18-395) showed a severe form of the disease – classified as N3 stage – which is characterized by high levels of neoplastic cells infiltrating the gills, different levels of infiltration of the digestive gland and gonad, and low/very low infiltration of the mantle and foot (*Figure 1d,e*; *Figure 1—figure supplement 1*); one specimen (EMVV18-400) was found that was affected with an intermediate form

**eLife digest** In humans and other animals, cancer cells divide excessively, forming tumours or flooding the blood, but they rarely spread to other individuals. However, some animals, including dogs, Tasmanian devils and bivalve molluscs like clams, cockles and mussels, can develop cancers that are transmitted from one individual to another. Despite these cancers being contagious, each one originates in a single animal, meaning that even when the cancer has spread to many individuals, its origins can be traced through its DNA.

Cancer contagion is rare, but transmissible cancers seem to be particularly common in the oceans. In fact, 7 types of contagious cancer have been described in bivalve species so far. These cancers are known as 'hemic neoplasias', and are characterized by the uncontrolled division of blood-like cells, which can be released by the host they developed in, and survive in ocean water. When these cells encounter individuals from the same species, they can infect them, causing them to develop hemic neoplasia too

There are still many unanswered questions about contagious cancers in bivalves. For example, how many species do the cancers affect, and which species do the cancers originate in? To address these questions, Garcia-Souto, Bruzos, Díaz et al. gathered over 400 specimens of a species of clam called the warty venus clam from the coastlines of Europe and examined them for signs of cancer. Clams collected in two regions of Spain showed signs of hemic neoplasia: one of the populations was from the Balearic Islands in the Mediterranean Sea, while the other came from the Atlantic coast of northwestern Spain.

Analyzing the genomes of the tumours from each population showed that the cancer cells from both regions had likely originated in the same animal, indicating that the cancer is contagious and had spread through different populations. The analysis also revealed that the cancer did not originally develop in warty venus clams: the cancer cells contained DNA from both warty venus clams and another species called striped venus clams. These two species live close together in the Mediterranean Sea, suggesting that the cancer started in a striped venus clam and then spread to a warty venus clam. To determine whether the cancer still affected both species, Garcia-Souto, Bruzos, Díaz et al. screened 200 striped venus clams from the same areas, but no signs of cancer were found in these clams. This suggests that currently the cancer only affects the warty venus clam.

These findings confirm that contagious cancers can jump between clam species, which could be threat to the marine environment. The fact that the cancer was so similar in clams from the Atlantic coast and from the Mediterranean Sea, however, suggests that it may have emerged very recently, or that human activity helped it to spread from one place to another. If the latter is the case, it may be possible to prevent further spread of these sea-borne cancers through human intervention.

of the disease – N2 stage – characterized by low levels of neoplastic cells infiltrating the gill vessels, digestive gland, and gonad, but not the foot (*Figure 1—figure supplement 1*); and three specimens (ERVV17-2997, EMVV18-373, and EMVV18-376) were diagnosed with a light form of the disease – N1 stage – characterized by low levels of neoplastic cells infiltrating the gills vessels only, and no infiltration in the remaining tissues (*Figure 1—figure supplement 1*). Electron microscopy analysis through gill's ultrathin sections from two neoplastic warty venus specimens (ERVV17-2995 and ERVV17-3193) revealed tumour cells with a round shape and a pleomorphic nucleus, which are morphological features that generally characterize bivalves' HN (*Figure 1f*; *Figure 1—figure supplement 2*). Finally, one additional neoplastic warty venus specimen (EVVV11-02) was included in the study. The animal, which was sampled in 2011 in Galicia and came from a private collection, showed abnormal metaphases in the gills that were suggestive of HN. Although the species typically shows a 2n = 38 karyotype with metacentric chromosomes that are homogeneous in size (*García-Souto et al., 2015*), the tumoural metaphases from this individual showed around 100 chromosomes that were variable in size and shape (*Figure 1g*).

To obtain some biological insights into the clonal dynamics of this cancer, we carried out whole-genome sequencing with Illumina paired-ends in DNA samples isolated from the tumoural haemolymph from eight out of nine neoplastic specimens mentioned above (*Table 1*). Their feet were also sequenced, as foot typically represents the tissue with lower infiltration of neoplastic cells, making

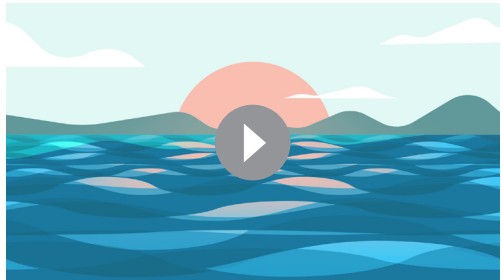

**Video 1.** Mitochondrial genome sequencing of marine leukaemias reveals cancer contagion between clam species in the Seas of Southern Europe. Infographic video outlining the main findings of the research carried out.

https://elifesciences.org/articles/66946/figures#video1

it a good candidate tissue to act as 'matched-normal' (i.e. host tissue). As for the animal with an abnormal karyotype (EVVV11-02) that was compatible with HN, we sequenced the only tissue available, which were gills (*Table 1*). Only one neoplastic specimen (EMVV18-373) that had a very low proportion of tumour cells in its haemolymph was excluded from the sequencing. Then, we mapped the paired-end reads onto a dataset containing non-redundant mitochondrial Cytochrome C Oxidase subunit 1 (*Cox1*) gene references from 118 Venerid clam species. In six out of eight sequenced neoplastic specimens, the results revealed an overrepresentation (>99%) of reads in the sequenced tissues mapping to *Cox1* DNA sequences that exclusively identified two different clam species (*Figure 2a*): the expected one, warty venus clam (*V. verrucosa*), and a second, unexpected one, the striped venus (*C. gallina*), a clam that inhabits the Mediterranean Sea (*Figure 2b*). Preliminary analysis by PCR and capillary sequencing of *Cox1* in the haemolymph of two

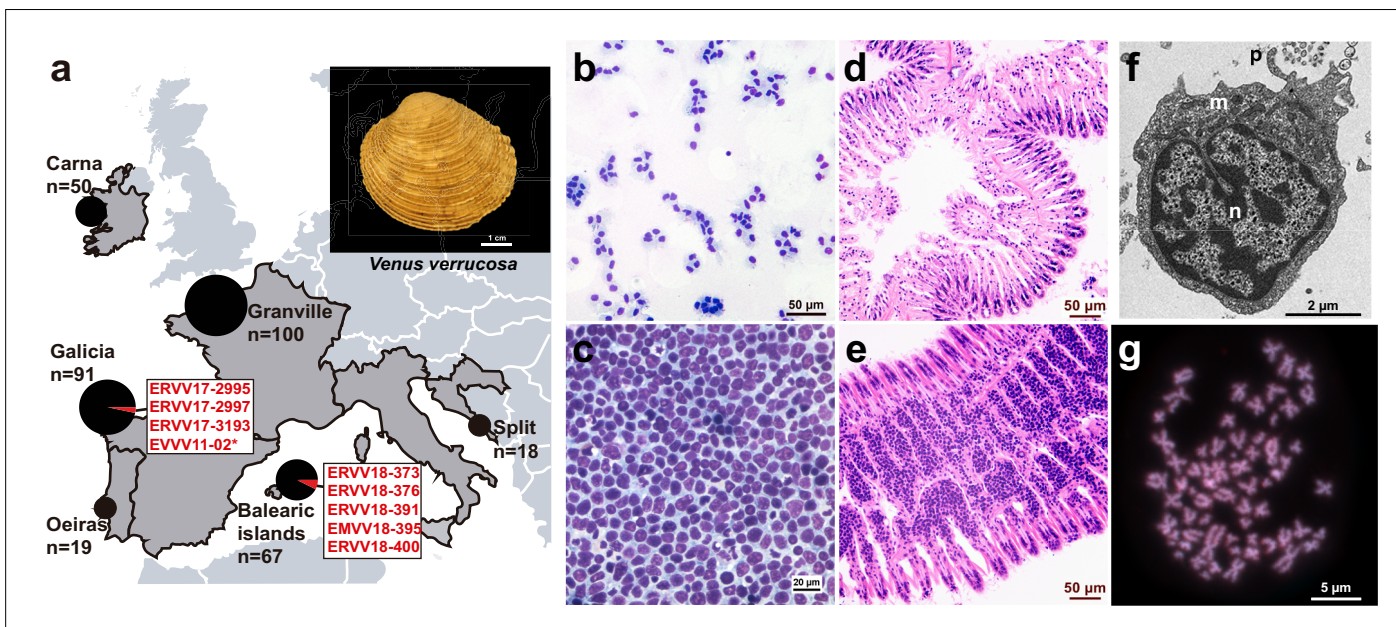

**Figure 1.** Geographical location of warty venus (*V. verrucosa*) specimens and diagnosis of hemic neoplasia. (**a**) Locations of *V. verrucosa* clams collected for this study and specimens diagnosed with hemic neoplasia. Size of the pie charts correlates with the number of samples collected (number of samples '*n*' is shown together with each pie chart). Pie charts show the proportion of samples with hemic neoplasia (black, no neoplastic specimens; red, neoplastic specimens). Codes of neoplastic samples are shown. Top-right corner shows a representative specimen of the species *V. verrucosa*. (**b**) Cytological examination of haemolymph smear (Hemacolor stain) from a healthy (N0) specimen, ERVV17-2963, shows normal haemocytes. (**c**) Haemolymph smear of a *V. verrucosa* specimen with high-grade (N3 stage) hemic neoplasia, ERVV17-3193, shows neoplastic cells that replaced normal haemocytes. (**d**) Detail of haematoxylin and eosin-stained of histological section from the gills of the healthy (N0) specimen ERVV17-2990. (**e**) Same for ERVV17-2995, a specimen infected with a high-grade (N3 stage) hemic neoplasia, showing neoplastic cells infiltrating the gills. (**f**) Transmission electron microscopy analysis of a *V. verrucosa* hemic neoplasia tumour cell shows a round shape, pseudopodia 'p', pleomorphic nucleus 'n' with scattered heterochromatin, and mitochondria 'm'. (**g**) Metaphase chromosomes from a neoplastic cell found in the gills of the *V. verrucosa* specimen EVVV11-02, showing abnormal chromosome number (>19 pairs) and abnormal chromosome morphology. Chromosomes stained with 4',6-DiAmidino-2-PhenylIndole (DAPI) and Propidium Iodide (PI).

The online version of this article includes the following figure supplement(s) for figure 1:

**Figure supplement 1.** Histological diagnosis of hemic neoplasia in warty venus (*V. verrucosa*) specimens.

**Figure supplement 2.** Transmission electron microscopy (TEM) of healthy and neoplastic *V. verrucosa* specimens.

**Table 1.** Clam specimens and tissues sequenced with Illumina paired-ends.

Sixteen specimens (eight neoplastic and eight non-neoplastic) from three different clam species (*V. verrucosa*, *C. gallina*, and *C. striatula*) were sequenced with Illumina paired-ends. Columns 5 and 6 show the number of reads generated for the host tissue (when neoplastic, matched-normal tissue was foot) and the tumoural haemolymph, respectively. (*) denotes the only available tissue from this neoplastic animal, collected in 2011, were gills. (#) denotes hemic neoplasia stage was not determined because cytohistological examination was not possible in this individual, which was diagnosed by cytogenetics.

| Clam species | Specimen origin | Specimen code | Diagnosis | Foot reads | Haemolymph reads |
|---|---|---|---|---|---|
| *V. verrucosa* | Galicia, Spain | ERVV17-2995 | N3 | 833 M | 919 M |
| *V. verrucosa* | Galicia, Spain | ERVV17-2997 | N1 | 766 M | 598 M |
| *V. verrucosa* | Galicia, Spain | ERVV17-3193 | N3 | 739 M | 850 M |
| *V. verrucosa* | Balearic Islands, Spain | EMVV18-376 | N1 | 784 M | 849 M |
| *V. verrucosa* | Balearic Islands, Spain | EMVV18-391 | N3 | 617 M | 623 M |
| *V. verrucosa* | Balearic Islands, Spain | EMVV18-395 | N3 | 697 M | 679 M |
| *V. verrucosa* | Balearic Islands, Spain | EMVV18-400 | N1 | 782 M | 1133 M |
| *V. verrucosa* | Galicia, Spain | EVVV11-02 | N# | 743 M* | –* |
| *V. verrucosa* | Split, Croatia | CSVV18-1052 | Healthy | 161 M | – |
| *V. verrucosa* | Balearic Islands, Spain | EMVV18-385 | Healthy | 143 M | – |
| *V. verrucosa* | Granville, France | FGVV18-183 | Healthy | 752 M | – |
| *V. verrucosa* | Carna, Ireland | IGVV19-666 | Healthy | 155 M | – |
| *V. verrucosa* | Oeiras, Portugal | PLVV18-2249 | Healthy | 163 M | – |
| *C. gallina* | S.Benedetto, Italy | IMCG15-69 | Healthy | 147 M | – |
| *C. gallina* | Cadiz, Spain | ECCG15-201 | Healthy | 752 M | – |
| *C. striatula* | Galicia, Spain | EVCS14-09 | Healthy | 706 M | – |

neoplastic specimens, EMVV18-373 and EVVV11-02, revealed an electropherogram with overlapping peaks apparently containing two different haplotypes that match the reference *Cox1* sequences for warty and striped venus (*Figure 2c*).

These results suggested cancer contagion between the two clam species of the family *Veneridae*. Hence, to decipher the origins of this clam neoplasia, we further analysed the mitochondrial DNA (mtDNA) from the two species involved and the tumours. Firstly, we performed multiplatform genome sequencing, including Illumina short reads and Oxford Nanopore long reads, on canonical individuals from the two species to obtain a preliminary assembly of the mitogenomes of *V. verrucosa* and *C. gallina*. These reconstructions resulted in 18,092- and 17,618-bp long mtDNA genomes for the warty venus and the striped venus clam, respectively (*Figure 2—figure supplement 1*). The comparative analysis of the nucleotide sequences from both mitogenomes confirms that, although both species are relatively close within the subfamily *Venerinae* (*Canapa et al., 1996*), they represent distinct sister species, showing a Kimura's two-parameter nucleotide distance (K2P) equal to 21.13%. Then, we mapped the paired-end sequencing data from the six neoplastic specimens with evidence of interspecies cancer transmission onto the two reconstructed species-specific mtDNA genomes. This approach confirmed the coexistence of two different mtDNA haplotypes in the six examined neoplastic samples, matching the canonical mtDNA genomes from the two clam species. For example, in a N2-stage specimen (EMVV18-400), this analysis revealed different proportion of tumour and host mtDNA molecules in the two tissue types sequenced (*Figure 2d*). Here, the striped venus mtDNA results the most abundant in the haemolymph, in which tumour cells are dominant over the remaining cell types, and the lower in the matched-normal tissue (i.e. infiltrated foot), where tumour cells represent a minor fraction

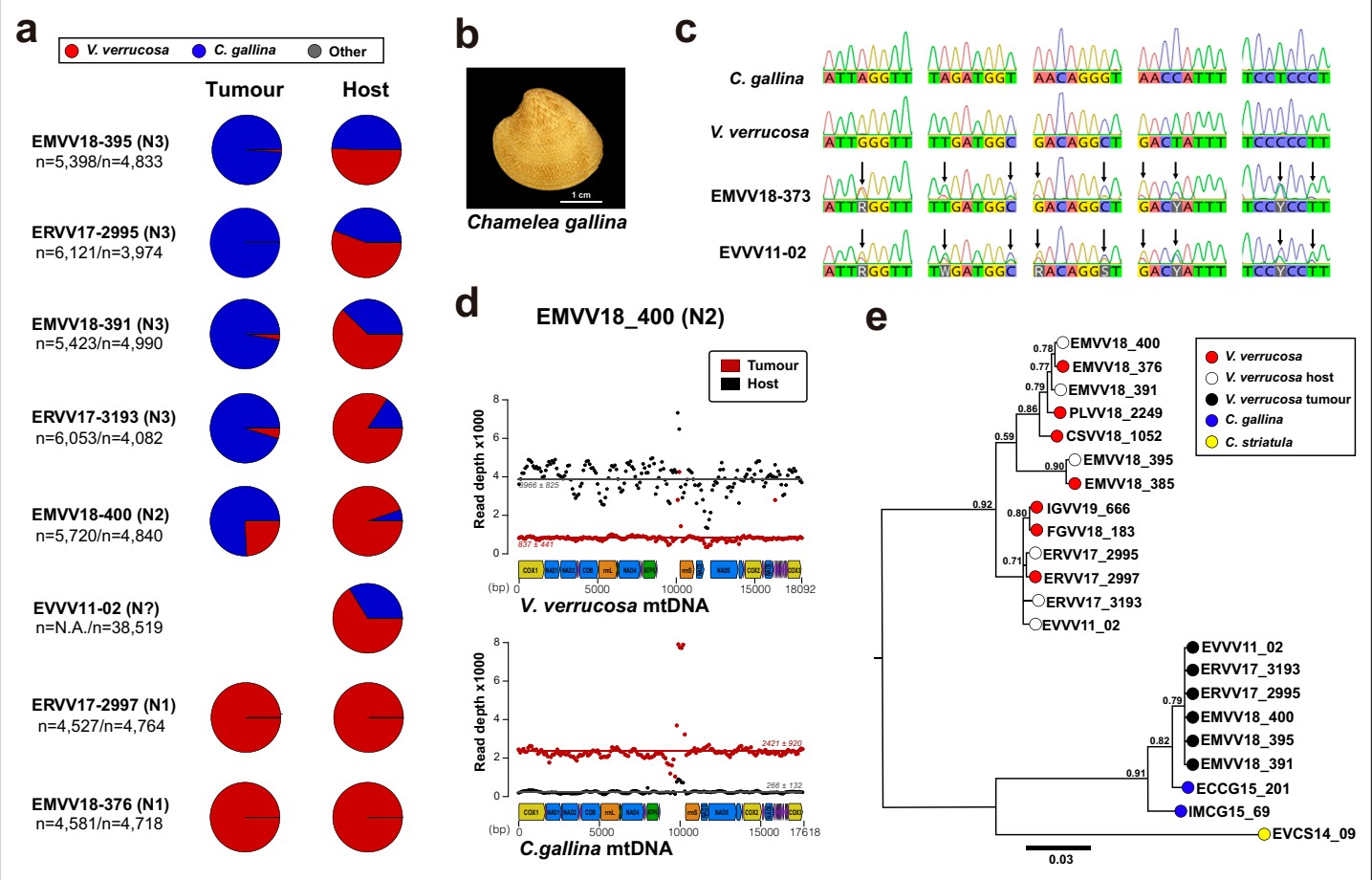

**Figure 2.** Mitochondrial DNA sequencing and phylogenetic analyses reveal cancer contagion between warty venus (*V. verrucosa*) and striped venus (*C. gallina*) clam species. (**a**) In eight warty venus specimens sequenced with Illumina paired-ends, the pie charts show the proportion of reads mapping *Cox1* reference sequences from 137 different *Verenidae* species, including *V. verrucosa* (red), *C. gallina* (blue), and the remaining species (grey). Two different tissues were sequenced: the tumour tissue (left pie chart), typically haemolymph, and the host/matched-normal tissue (right pie chart), typically foot. Note that for specimen EVVV11-02 only the host/matched-normal tissue (gills) was available. '*n*' denotes the total number of reads mapping the *Cox1* reference for the tumour tissue (left), and the host tissue (right). (**b**) Representative specimen of the species *C. gallina*. (**c**) Capillary sequencing electropherograms of mitochondrial *Cox1* gene fragments from two neoplastic *V. verrucosa* specimens (EMVV18-373 and EVVV11-02) and two healthy reference specimens from *V. verrucosa* and *C. gallina*. The results show overlapping peaks (arrows) in the sequenced tissues from the neoplastic animals, which suggest coexistence of mitochondrial DNA (mtDNA) haplotypes from two clam species. (**d**) In *V. verrucosa* neoplastic (N2-stage) specimen EMVV18-400, mtDNA read depth shows different proportion of warty venus and striped venus mtDNA haplotypes in the tumour tissue (haemolymph) and the matched-normal tissue (foot). (**e**) Molecular phylogeny using Bayesian inference inferred on the alignment of all mitochondria coding genes and rRNA gene sequences (15 loci) that includes six neoplastic *V. verrucosa* specimens with evidence of cancer contagion from *C. gallina*. Bootstrap values are shown above the branches.

The online version of this article includes the following figure supplement(s) for figure 2:

**Figure supplement 1.** Draft reference mitochondrial DNA (mtDNA) genome assemblies reconstructed for *V. verrucosa*, *C. gallina*, and *C. striatula*.

**Figure supplement 2.** Read depth analysis of the mitochondrial DNA (mtDNA) confirms coexistence of two different clam species haplotypes.

of the total. Similar results were obtained for the remaining five neoplastic individuals (*Figure 2— figure supplement 2*).

To further investigate the evolutionary origins and geographic spread of this cancer, we sequenced with Illumina paired-ends an additional set of eight healthy (i.e. non-neoplastic) clams from three different *Veneridae* species, including five more warty venus specimens (EMVV18-385, IGVV19-666, FGVV18-183, CSVV18-1052, and PLVV18-2249) from five different countries, two striped venus specimens (IMCG15-69 and ECCG15-201) from two countries, and one specimen (EVCS14-09) from its sibling species *Chamelea striatula*, a type of striped venus clam that inhabits the Atlantic Ocean from Norway to the Gulf of Cadiz in Spain. This made a total of 16 *Veneridae* specimens sequenced,

all listed in *Table 1* (see also *Supplementary file 1*). The complete mitochondrial genomes from all tumoural and healthy *V. verrucosa* specimens (13 individuals), 2 *C. gallina*, and 1 from its sibling species *C. striatula*, were individually de novo assembled from the sequencing reads. As expected, this approach reconstructed two different haplotypes in six out eight sequenced neoplastic animals, supporting the presence of mtDNA from two different species. Despite the high sequencing coverage obtained for these individuals (*Table 1*), we did not find foreign reads in the N1 tumours (ERVV17-2997 and EMVV18-373), most likely due to a low proportion of neoplastic cells in the haemolymph and the matched-normal tissue. Then, we performed a phylogenetic analysis based on the alignment of these mitochondrial genomes (13 coding and 2 RNA gene sequences, altogether encompassing ~14 kb). The results show that tumour and non-tumour sequences from neoplastic warty venus specimens define two well-differentiated clades, and that tumoural warty venus sequences are all identical and closer to striped venus mtDNA than to its own (warty venus) (*Figure 2e*). Overall, these data support the existence of a single cancer clone originated in the striped venus clam *C. gallina* that was transmitted to *V. verrucosa*.

Transmissible cancers are known to occasionally acquire mitochondria from transient hosts (*Strakova et al., 2016*; *Strakova et al., 2020*), which can lead to misinterpretation of their evolutionary history. Thus, we looked for nuclear markers to confirm the striped venus origin of this cancer lineage. We performed a preliminary draft assembly of the warty venus and the striped venus nuclear 'reference' genomes, using the paired-end sequencing data from two non-neoplastic animals. Then, we used bioinformatic approaches to find single copy nuclear genes that were homologous between the two species, identifying two confident candidate genomic regions (see Methods): a 2.9-kb long region from *DEAH12*, a gene that encodes for an ATP-dependent RNA helicase, and a 2.2-kb long fragment from the Transcription Factor II Human-like gene, *TFIIH*. With the idea of finding differentially fixed single-nucleotide variants (SNVs) between both species, we performed PCR amplification and capillary sequencing on a 441 bp fragment from the *DEAH12*, and a 559 bp fragment from *TFIIH*, in 2 cohorts of non-neoplastic warty venus specimens (12 for *DEAH12* and 15 for *TFIIH*), 2 cohorts of non-neoplastic striped venus (9 for *DEAH12* and 12 for *TFIIH*), and 1 specimen of its sister species *C. striatula*. This analysis provided 14 and 15 sites, respectively, for the *DEAH12* and the *TFIIH* loci, with fixed SNVs (allele frequency >95%) that allowed to discriminate between the 3 relevant species and the tumour (*Figure 3a*). These variants were employed to identify the Illumina reads from each sequenced warty venus neoplastic specimens that were specific for either warty venus or striped venus, which allowed to obtain the consensus sequences that corresponded to the tumour tissue and the non-affected tissue from each neoplastic individual. At the end of this process, we performed Maximum Likelihood phylogenetic reconstructions from these individual nuclear consensus sequences. On the one hand, the phylogeny for the *DEAH12* locus confirmed both the monophyly of the tumoural sequences and their closer relationship to *C. gallina* than to the host species (*Figure 3b*), which were also observed in the mtDNA analysis. However, the phylogeny derived from the *TFIIH* locus showed that, although the tumours remained monophyletic, they were positioned in a basal branch relative to *C. gallina* and *V. verrucosa* (*Figure 3b*). Hence, to resolve these differences we also obtained a multilocus species tree based on the alignment of both the mtDNA and the two nuclear genes. This new phylogeny confirmed that warty venus tumours are closer to striped venus specimens than to non-neoplastic warty venus sequences from the same diseased specimens, while the non-neoplastic sequences conformed a more distant warty venus lineage (*Figure 3c*).

To obtain further evidence on the striped venus origin of this clam's neoplasia, we performed a comparative screening of tandem repeats in the genomes of *C. gallina* and *V. verrucosa* using fluorescence in situ hybridization (FISH) (*Figure 3d*; *Figure 3—figure supplement 1*). We focused on two satellite DNA repeats, namely CL4 and CL17. The satellites represent repeats of 332- and 429-bp long monomers, respectively, and were identified in a preliminary bioinformatics screening of the striped venus reference genome (see methods). This FISH approach revealed that the mentioned repeats are very abundant in heterochromatic regions from the genomes of the canonical striped venus and the neoplastic warty venus specimens tested (*Figure 3d*; *Figure 3—figure supplement 1*). However, the repeats were absent in the metaphases from all the healthy warty venus individuals (*Figure 3d*; *Figure 3—figure supplement 1*). These results suggest that the relevant chromosomes with CL4 and CL17 satellites found in neoplastic warty venus specimens derive from *C. gallina*, supporting that a tumour originated in *C. gallina* was transmitted to *V. verrucosa*.

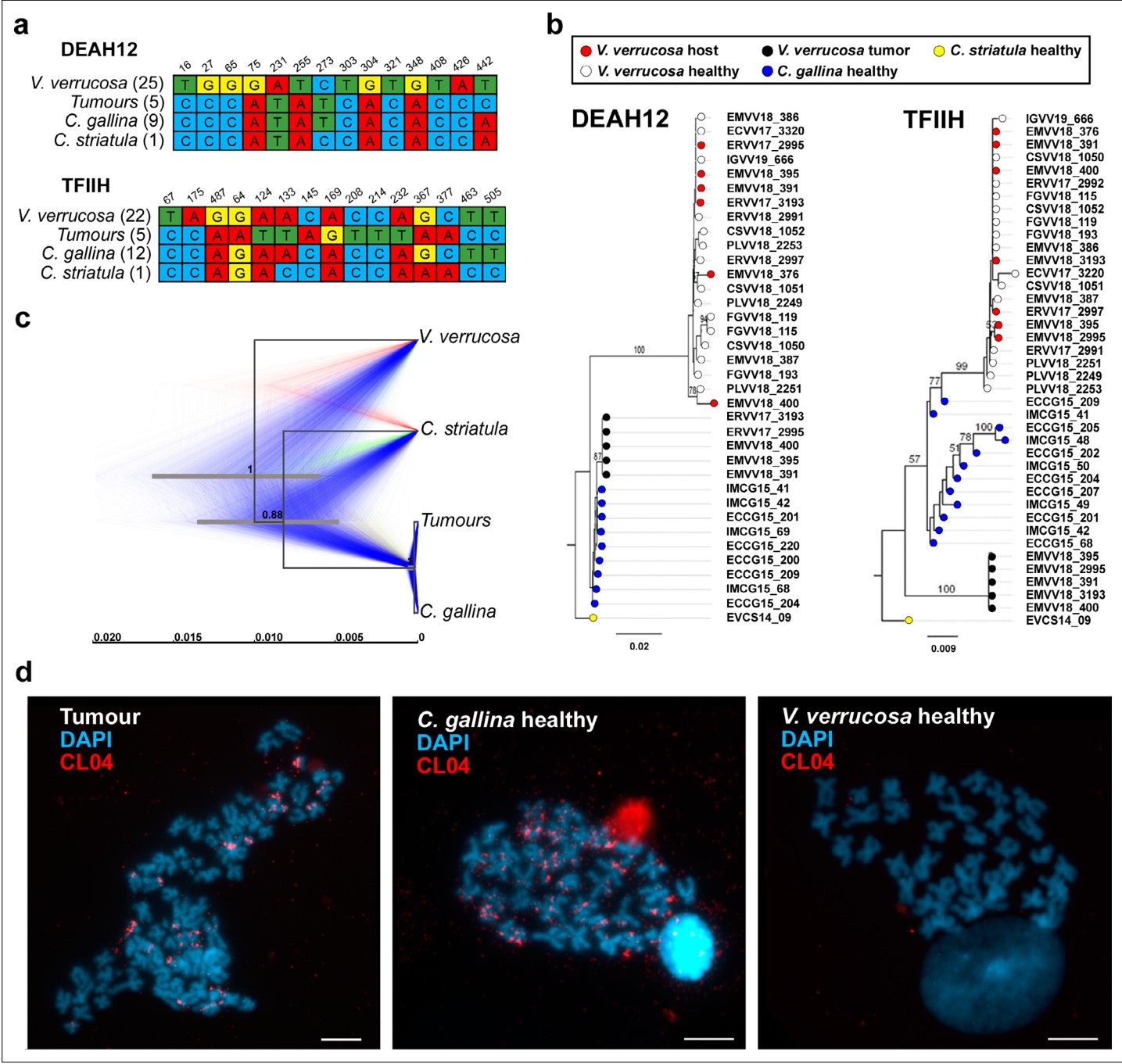

**Figure 3.** Nuclear DNA sequencing and phylogenetic analyses confirm a single cancer lineage spreading in populations of the warty venus (*V. verrucosa*) that originated in the striped venus (*C. gallina*). (**a**) Single-nucleotide variants discriminating between V. *verrucosa* tumours and the three canonical species (*V. verrucosa*, *C. gallina*, and *C. striatula*) along a 441- and a 559-bp long fragments of nuclear genes *DEAH12* and *TFIIH*, respectively. (**b**) Maximum Likelihood molecular phylogenies based on the two fragments of the nuclear DNA markers *DEAH12* and *TFIIH*. Bootstrap support values (500 replicates) from Maximum Likelihood analyses above 50 are shown on the corresponding branches. (**c**) Multispecies coalescent (MSC) tree of *V. verrucosa*, their tumours and *Chamelea* sp. based on the entire mitochondrial DNA (mtDNA) and the two nuclear markers, *DEAH12* and *TFIIH*. A maximum clade credibility (MCC) tree is shown, with posterior probabilities below the branches, and 95% highest probability density (HPD) intervals of node heights as grey bars. The trees distribution shown includes 1000 trees and represents the range of alternative topologies, in which blue is the most common set of topologies, red the second most common one, and green the remaining. (**d**) Fluorescence in situ hybridization (FISH) to specifically detect the satellite DNA CL4 in one *V. verrucosa* tumour and healthy specimens from the species *C. gallina* and *V. verrucosa* shows probes accumulate in heterochromatic regions, mainly in subcentromeric and subtelomeric positions, from the chromosomes of the tumour and the healthy *C. gallina* tested but not in healthy *V. verrucosa*.

*Figure 3 continued on next page*

*Figure 3 continued*

The online version of this article includes the following figure supplement(s) for figure 3:

**Figure supplement 1.** CL17 satellite DNA supports tumour transmission from *C. gallina*.

To find out whether this cancer is present in the clam species where it first arose, we performed a screening for its presence in natural populations of striped venus clams from the species *C. gallina* (*n* = 213) and *C. striatula* (*n* = 9) at five additional sampling points across two countries (*Supplementary file 1*), including Spain (*n* = 115) and Italy (*n* = 107). Histological analyses did not show any traces of HN in these specimens. The virtual absence of this tumour in natural populations of striped venus clams may suggest that today this leukaemia is being mainly, if not exclusively, transmitted between specimens of the recipient species, warty venus. However, further sampling in other regions across the striped venus area of distribution may be necessary to confirm these findings.

Overall, the results provided here reveal the existence of a transmissible leukaemia originated in a striped venus clam, most likely *C. gallina*, which was transmitted to a second species, the warty venus clam (*V. verrucosa*), and among whose specimens it currently propagates. We identified this parasitic cancer in warty venus clams from two sampling points that are more than 1000 nautical miles away in the coasts of Spain, bathed by two different seas, the Atlantic Ocean and the Mediterranean Sea. The analysis of mitochondrial and nuclear gene sequences revealed no nucleotide diversity within the seven tumours sequenced, which supports that all belong to the same neoplastic lineage that spreads between Veneridae clams in the Seas of Southern Europe. Although we ignore the age of this cancer clone, we can confirm it arose before 2011, when the neoplastic warty venus specimen EVVV11-02 was collected. The apparent lack of genetic variation between all tumours, even from distant sampling points, suggests either that this cancer is very recent, or that it may have been unintentionally scattered by the action of man, a way of transmission that has been proposed for other bivalve transmissible cancers (*Yonemitsu et al., 2019*).

## Materials and methods

### Sampling of clam specimens

We collected 570 clam specimens from three different species, from the following countries and locations (*Supplementary file 1*). *V. verrucosa* clams were collected in Spain (Galicia, *n* = 90; Balearic Islands, *n* = 67), France (Granville, *n* = 100), Croatia (Split, *n* = 18), Portugal (Oeiras, *n* = 19), and Ireland (Carna, *n* = 50). *C. gallina* clams were collected in Spain (Cadiz, *n* = 50; Mallorca, *n* = 50) and Italy (Naples, *n* = 50; Cattolica, *n* = 57). *C. striatula* clams were collected in Spain (Combarro, *n* = 9). Additionally, we recruited samples from the following specimens from private collections: one *V. verrucosa* clam collected in 2011 in Spain (Islas Cies), four *C. gallina* collected in 2015 in Italy (San Benedetto de Tronto), five *C. gallina* collected in 2015 in Spain (Huelva), and one *C. striatula* collected in 2014 in Spain (Marin).

### Diagnosis of HN

We followed standard cytological and/or histological protocols to test and diagnose HN in the clam specimens. However, only histological examination resulted decisive for the diagnosis, particularly in early stages of the disease. Briefly, for each animal, we extracted 300–2000 ml of haemolymph from the posterior adductor muscle using a 5 ml syringe with a 23 G needle. The haemolymph (50 ml) was diluted in cold Alserver's antiaggregant solution to a 1:4 concentration, and spotted by centrifugation (130 × *g*, 4°C, 7 min) onto a microscope slide using cytology funnel sample chambers to produce a cell monolayer. Haemolymph smears were fixed and stained with Hemacolor solutions from Sigma-Aldrich and subsequently examined with a light microscope for the diagnosis of HN. Tissues (visceral mass, gills, mantle, and foot) were dissected, fixed in Davidson's solution and embedded in paraffin. Then, 5-mm thick sections from each tissue were microdissected and stained with Harris' haematoxylin and eosin and examined using a light microscope for histopathological analysis. HN was diagnosed and classified according to three disease stages (i.e. N1, N2, or N3) as follows. N1 stage: small groups of leukaemic cells were detected only in the vessels of the gills and in the connective tissue surrounding the digestive tubules. N2 stage: leukaemic cells spread to different organs, conforming small groups

in the connective tissue that surrounds the digestive gland and the gonadal follicles, branchial sinuses, and mantle. N3 stage: leukaemic cells invade the filaments, completely deforming the plica structure in the gill, invade the connective tissue surrounding the gonadal follicles and the digestive gland; in the mantle, they invade the connective tissue, but in the muscle fibres of the mantle and foot, cells appear isolated or in small groups and in lower intensity than in other tissues.

## Electron microscopy analysis

Four *V. verrucosa* specimens (two non-neoplastic, ERVV17-2993 and ERVV17-2992, and two with high grade of HN, ERVV17-2995 and ERVV17-3193) were processed for transmission electron microscopy as follows: 2 mm sections of gills and digestive glands were fixed in 2.5% glutaraldehyde seawater for 2 hr at 4°C. Then, tissues were post-fixed in 1% osmium tetroxide in sodium cacodylate solution and embedded in Epon resin. Ultrathin sections were stained with uranyl acetate and lead citrate and examined in a JEM-1010 transmission electron microscope.

## Cytogenetics

Mitotic chromosomes of a neoplastic *V. verrucosa* specimen (EVVV11-02) were obtained as follows. After colchicine treatment (0.005%, 10 hr), gills were dissected, treated with a hypotonic solution, and fixed with ethanol and acetic acid. Small pieces of fixed gills were disaggregated with 60% acetic acid to obtain cell suspensions that were spread onto preheated slides. Chromosome preparations were stained with DAPI (0.14 mg/ml) and PI (0.07 mg/ml) for 8 min, mounted with antifade medium, and photographed. A comparative screening of tandem repeats was performed on the genomes of *C. gallina* and *V. verrucosa* using RepeatExplorer (*Novák et al., 2010*) on a merged short-read dataset of both species (500,000 reads each). Short reads of healthy and neoplastic animals were mapped onto both satellite consensus sequences using BWA, filtered according to their mapping quality (q > 60 and AS >70) and their abundance assessed by means of samtools/bamtools. Satellites CL4 and CL17 were selected for FISH purposes and FISH probes were PCR amplified (CL4F: TCAGAAACCGCT ATTTTTCAC, CL4R: AAATGATGCTACGAACCTCC and CL17F: ATTCCAGAAATGTACATGAACAC, CL17R: ATTTTTGCACCAGATGTTCAC, respectively) and directly labelled with digoxigenin-11-dUTP (10× DIG Labeling Mix, Roche Applied Science). FISH experiments were performed as described in reference (*García-Souto et al., 2015*).

## De novo assembly of mitochondrial genomes and annotation

In total, we performed whole-genome sequencing on 23 samples from 16 clam specimens, which includes 8 neoplastic and 8 non-neoplastic animals by Illumina paired-end libraries of 350 bp insert size and reads 150 bp long. First we assembled the mitochondrial genomes of one *V. verrucosa* (FGVV18_193), one *C. gallina* (ECCG15_201), and one *C. striatula* (EVCS14_02) specimens with MITObim v1.9.1 (*Hahn et al., 2013*), using gene baits from the following *Cox1* and *16S* reference genes to prime the assembly of clam mitochondrial genomes: *V. verrucosa* (*Cox1*, with GenBank accession number KC429139; and *16S*: C429301), *C. gallina* (*Cox1*: KY547757, *16S*: KY547777), and *C. striatula* (*Cox1*: KY547747, *16S*: KY547767). These draft sequences were polished twice with Pilon v1.23 (*Walker et al., 2014*), and conflictive repetitive fragments from the mitochondrial control region were resolved using long read sequencing with Oxford Nanopore technologies (ONT) on a set of representative samples from each species and tumours. ONT reads were assembled with Miniasm v0.3 (*Li, 2016*) and corrected using Racon v1.3.1 (*Vaser et al., 2017*). Protein-coding genes, rDNAs and tDNAs were annotated on the curated mitochondrial genomes using MITOS2 web server (*Bernt et al., 2013*), and manually curated to fit ORFs as predicted by ORF-FINDER (*Rombel et al., 2002*). Then, we employed the entire mtDNAs of *V. verrucosa* (FGVV18_193) and *C. gallina* (ECCG15_201) as 'references' to map reads from individuals with neoplasia, filter reads matching either mitogenome and assemble and polish their two (healthy and tumoural) mitogenomes individually as above. Further healthy individuals were later sequenced and their mitogenomes assembled, to further investigate the geographic and taxonomic spread of this neoplasia.

## Analysis of *Cox1* sequences

We retrieved a dataset of 3745 sequences comprising all the barcode-identified venerid clam *Cox1* fragments available from the Barcode of Life Data System (BOLD, http://www.boldsystemns.org/).

Redundancy was removed using CD-HIT (*Fu et al., 2012*), applying a cut-off of 0.9 sequence identity, and sequences were trimmed to cover the same region. Whole-genome sequencing data from both healthy and tumoural warty venus clams were mapped onto this dataset, containing 118 venerid species-unique sequences, using BWA-mem, filtering out reads with mapping quality below 60 (-q60), and quantifying the overall coverage for each sequence with samtools idxstats. PCR primers were designed with Primer3 v2.3.7 (*Kõressaar et al., 2018*) to amplify a fragment of 354 bp from the *Cox1* mitochondrial gene of *V. verrucosa* and *C. gallina* (F: CCT ATA ATA ATT GGK GGA TTT GG, R: CCT ATA ATA ATT GGK GGA TTT GG). PCR products were purified with ExoSAP-IT and sequenced by Sanger sequencing.

## Mitochondrial genome coverage analysis

We further mapped the paired-end sequencing data from healthy and neoplastic tissues from all neoplastic samples onto the 'reference' mitochondrial genomes of *V. verrucosa* and *C. gallina* (two of the previously assembled ones, FGVV18_193 and ECCG15_201) using BWA-mem v0.7.17-r1188 (*Li and Durbin, 2009*) with default parameters. Duplicate reads were marked with Picard 2.18.14 and removed from the analysis. Read coverage depth was computed with samtools v1.9 (*Li et al., 2009*), summarized by computing the average in windows of 100 bp size and plotted with R v3.5.3.

## Draft assembly of nuclear reference genomes, identification of variable single copy orthologous nuclear loci and SNPs

We ran the MEGAHIT v1.1.3 assembler (*Li et al., 2015*) on the Illumina paired-end sequencing data to obtain partial nuclear genome assemblies of *V. verrucosa* (FGVV18_193), *C. gallina* (ECCG15_201), and *C. striatula* (EVCS14_02). Then, single copy genes were predicted with Busco v.3.0.2 (*Seppey et al., 2019*). Candidate genes were considered if they (1) were present in the genomes of the three species, and (2) showed variant allele frequencies (VAFs) at exclusively 0, 0.5, or 1.0 in all the sequenced healthy (non-neoplastic) specimens. Under this criteria, two loci were finally selected: a 3914-bp long fragment of *DEAH12*, a gene encoding for an ATP-dependent RNA helicase and a 2.2-kp length fragment of the Transcription Factor II Human-like gene, *TFIIH*. PCR primers were designed with Primer3 v2.3.7 to amplify and sequence a 441-bp region of the *DEAH12* nuclear gene (DEAH12_F: AGGT ATGCTGAAACAAACACTT and DEAH12_R: ACGACAAATTTGATACCTGGAAT) and a 559-bp fragment of the *TFIIH* gene (TFIIH_F: TGGCATCTTTGTTACGGAC and TFIIH_R: CTTGTGRTTCTGTATC TGATCAATAA) on neoplastic specimens from *V. verrucosa* and healthy animals from both species (*DEAH12*: 11 *V. verrucosa* and 9 *C. gallina*; *TFIIH*: 15 *V. verrucosa* and 12 *C. gallina*). We screened for differentially fixed SNVs between both species using the dapc function in the R package Exploratory Analysis of Genetic and Genomic Data adegenet (*Jombart and Ahmed, 2011*). These variants were later employed to filter the Illumina short reads matching either *V. verrucosa* or *C. gallina* genotypes from the neoplastic animals, and to obtain consensus sequences from tumour and healthy tissue in each sequenced specimen. Read filtering was performed with samtools fillmd, while GATK mutect2 (*Benjamin et al., 2019*) was used for variant calling. Only variants with VAFs close to fixation (>0.9) were considered when building the consensus sequences.

## Phylogenetic analyses

Mitochondrial sequences for 13 coding genes and 2 rDNA genes from the 23 recovered mitogenomes (6 neoplastic, 17 from host and healthy specimens) were extracted from the paired-end sequencing data by mapping reads onto the previously reconstructed canonical mtDNAs for *V. verrucosa* and *C. gallina* (*Figure 2—figure supplement 1*), concatenated, and subjected to multiple alignment with MUSCLE v3.8.425 (*Edgar, 2004*). The best-fit model of nucleotide substitution for each individual gene was selected using JModelTest2 (*Darriba et al., 2012*) and a partitioned Bayesian reconstruction of the phylogeny was performed with MrBayes v3.2.6 (*Ronquist et al., 2012*). Two independent Metropolis-coupled Markov Chain Monte Carlo (MCMC) analyses with four chains in each were performed. Each chain was run for 10 million generations, sampling trees every 1000 generations. Convergence of runs was assessed using Tracer (*Rambaut et al., 2018*).

*DEAH12* and *TFIIH* sequences were subjected to multiple alignment using MUSCLE v3.8.425. Then, a 'species/population tree' was inferred with the starBEAST multispecies coalescent model, as implemented in BEAST v2.6.2 (*Bouckaert et al., 2019*). This analysis was performed using a Yule

speciation prior and strict clock, with the best-fit model of nucleotide substitution obtained with jModelTest2 on both the concatenated mitochondrial haplotypes (13 protein-coding and 2 rRNAs genes) and unphased data from *DEAH12* and *TFIIH* nuclear fragments. The four mitochondrial groups observed on the mitogenome analysis (*V. verrucosa*, *C. gallina*, *C. striatula*, and Tumour) were defined as tips for the species tree. A single MCMC of 10 million iterations, with sampling every 1000 steps, was run. A burn-in of 10% was implemented to obtain ESS values above 200 with Tracer v1.7.1 and the resulting posterior distributions of trees were checked with DENSITREE v2.1 (*Bouckaert, 2010*). A maximum clade credibility tree was obtained with TreeAnnotator (*Bouckaert et al., 2019*) to summarize information on topology, with 10% burn-in and Common Ancestors for the node heights.

## Acknowledgements

We thank the Galicia Supercomputing Centre (CESGA) for the availability of informatic resources. JMCT, SR, SD, and JT are supported by European Research Council (ERC) Starting Grant 716,290 SCUBA CANCERS. ALB is supported by MINECO PhD fellowship BES-2016-078166. DG-S is supported by postdoctoral contract ED481B/2018/091 from Xunta de Galicia. DP is supported by ERC grant ERC-617457-PHYLOCANCER and by Spanish Ministry of Economy and Competitiveness (MINECO) grant PID2019-106247GB-I00. This research was partially funded by the European Union's Horizon 2020 research and innovation programme under grant agreement 730984, ASSEMBLE Plus project. CESAM got financial support from FCT/MEC (UIDP/50017/2020, UIDB/50017/2020).

## Additional information

### Funding

| Funder | Grant reference number | Author |
| --- | --- | --- |
| H2020 European Research Council | 716290 | Jose MC Tubio |
| MINECO PhD fellowship | BES-2016–078166 | Alicia L Bruzos |
| Xunta de Galicia | ED481B/2018/091 | Daniel Garcia-Souto |
| European Research Council | 617457 | David Posada |
| Spanish Ministry of Economy and Competitiveness (MINECO) | PID2019-106247GB-I00 | David Posada |
| European Union's Horizon 2020 research and innovation programme | 730984 | Seila Diaz |
| CESAM | UIDP/50017/2020 + UIDB/50017/2020 | Fernando Ricardo |

The funders had no role in study design, data collection, and interpretation, or the decision to submit the work for publication.

### Author contributions

Daniel Garcia-Souto, Conceptualization, Data curation, Formal analysis, Funding acquisition, Investigation, Methodology, Project administration, Resources, Supervision, Validation, Writing – original draft, Writing – review and editing; Alicia L Bruzos, Conceptualization, Data curation, Formal analysis, Investigation, Methodology, Project administration, Validation, Writing – original draft; Seila Diaz, Conceptualization, Data curation, Formal analysis, Funding acquisition, Investigation, Methodology, Project administration, Resources, Supervision, Validation, Writing – original draft; Sara Rocha, Conceptualization, Data curation, Formal analysis, Funding acquisition, Investigation, Methodology, Project administration, Resources, Software, Supervision, Validation, Writing – review and editing; Ana Pequeño-Valtierra, Rosana Rodriguez, Jorge Rodriguez-Castro, Jose MC Tubio, Conceptualization,

Data curation, Formal analysis, Funding acquisition, Investigation, Methodology, Project administration, Resources, Software, Supervision, Validation, Writing – original draft, Writing – review and editing; Camila F Roman-Lewis, Juana Alonso, Investigation, Methodology; Damian Costas, Antonio Villanueva, Methodology, Project administration, Resources; Luis Silva, Methodology, Project administration, Resources, Writing – review and editing; Jose Maria Valencia, Giovanni Annona, Andrea Tarallo, Fernando Ricardo, Ana Bratoš Cetinić, Resources, Writing – review and editing; David Posada, Investigation, Methodology, Writing – review and editing; Juan Jose Pasantes, Investigation, Methodology, Resources, Supervision, Writing – review and editing

### Author ORCIDs
Daniel Garcia-Souto http://orcid.org/0000-0002-0997-8799
Alicia L Bruzos http://orcid.org/0000-0003-4362-545X
Seila Diaz http://orcid.org/0000-0002-6607-290X
Sara Rocha http://orcid.org/0000-0002-9705-1511
Jorge Rodriguez-Castro http://orcid.org/0000-0003-2912-9601
Jose Maria Valencia http://orcid.org/0000-0002-1912-1975
Giovanni Annona http://orcid.org/0000-0001-7806-6761
Ana Bratoš Cetinić http://orcid.org/0000-0002-2928-4858
David Posada http://orcid.org/0000-0003-1407-3406
Jose MC Tubio http://orcid.org/0000-0003-3540-2459

### Ethics
No particular ethical aspects are necessary for bivalves research in the laboratory.

### Decision letter and Author response
Decision letter https://doi.org/10.7554/eLife.66946.sa1
Author response https://doi.org/10.7554/eLife.66946.sa2

## Additional files

### Supplementary files
• Supplementary file 1. Sampling data of 570 specimens analysed in this study.

• Transparent reporting form

### Data availability
Nucleotide data for the mitochondrial DNA assemblies has been uploaded to GenBank under accession numbers MW662590-MW662611. These correspond to reference healthy animals of Venus verrucosa (MW662590, MW662593, MW662607, MW662608 and MW662610), Chamelea gallina (MW662591 and MW662609), Chamelea striatula (MW662611) and to both normal (MW662592, MW662595, MW662597, MW662599, MW662601, MW662602, MW662604, MW662606) and tumoral (MW662594, MW662596, MW662598, MW662600, MW662603, MW662605) mitochondrial DNAs of neoplastic animals. Genomic data (Illumina paired-end sequencing) from the mitogenomes analysed in this manuscript are allocated in DRYAD (https://doi.org/10.5061/dryad.zcrjdfn9v).

The following dataset was generated:

| Author(s) | Year | Dataset title | Dataset URL | Database and Identifier |
|---|---|---|---|---|
| Garcia-Souto D, Diaz-Costas S, Bruzos A, Rocha S, Roman-Lewis C, Alonso J, Rodriguez R, Jorge R, Villanueva A, Silva L, Valencia J, Annona G, Tarallo A, Ricardo F, Bratos-Cetinic A, Posada D, Pasantes J, Tubio J MC | 2021 | Mitochondrial genome sequencing of marine leukemias reveals cancer contagion between clam species in the Seas of Southern Europe | https://doi.org/10.5061/dryad.zcrjdfn9v | Dryad Digital Repository, 10.5061/dryad.zcrjdfn9v |

*Continued*

| Author(s) | Year | Dataset title | Dataset URL | Database and Identifier |
|---|---|---|---|---|
| Garcia-Souto D, Diaz-Costas S, Bruzo AL, Rocha S, Pequeno A, Roman-Lewis CF, Costas D, Rodriguez-Castro J, Villanueva A, Silva L, Valencia JM, Annona G, Tarallo A, Richardo F, Bratos-Centinic A, Posada D, Tubio JM | 2022 | Venus verrucosa mitochondrion, complete genome CSVV18_1050 Healthy animal, foot | https://www.ncbi.nlm.nih.gov/nuccore/MW662590 | NCBI GenBank, MW662590 |
| Garcia-Souto D, Diaz-Costas S, Bruzos AL, Rocha S, Pequeno A, Roman-Lewis CF, Costas D, Rodriguez-Castro J, Villanueva A, Silva L, Valencia JM, Annona G, Tarallo A, Ricardo F, Bratos-Cetinic A, Tubio JM | 2022 | Venus verrucosa mitochondrion, complete genome EMVV18_385 Healthy animal, foot | https://www.ncbi.nlm.nih.gov/nuccore/MW662593 | NCBI GenBank, MW662593 |
| Garcia-Souto D, Diaz-Costas S, Bruzos AL, Rocha S, Pequeno A, Roman-Lewis CF, Costas D, Rodriguez-Castro J, Villanueva A, Silva L, Valencia JM, Tarallo A, Richardo F, Bratos-Cetinic A, Posada D, Tubio JM | 2022 | Venus verrucosa mitochondrion, complete genome FGVV18_193 Healthy animal, foot | https://www.ncbi.nlm.nih.gov/nuccore/MW662607 | NCBI GenBank, MW662607 |
| Garcia-Souto D, Diaz-Costas S, Bruzos AL, Rocha S, Pequeno A, Roman-Lewis CF, Costas D, Rodriguez-Castro J, Villanueva A, Silva L, Valencia JM, Annona G, Tarallo A, Richardo F, Bratos-Cetinic A, Posada D, Tubio JM | 2022 | Venus verrucosa mitochondrion, complete genome IGVV19_666 Healthy animal, foot | https://www.ncbi.nlm.nih.gov/nuccore/MW662608 | NCBI GenBank, MW662608 |
| Garcia-Souto D, Diaz-Costas S, Bruzos AL, Rocha S, Pequeno A, Roman-Lewis CF, Costas D, Rodriguez-Castro J, Villanueva A, Silva L, Valencia JM, Annona G, Tarallo A, Richardo F, Bratos-Cetinic A, Posada D, Tubio JM | 2022 | Venus verrucosa mitochondrion, complete genome PLVV18_2249 Healthy animal, foot | https://www.ncbi.nlm.nih.gov/nuccore/MW662610 | NCBI GenBank, MW662610 |

*Continued on next page*

*Continued*

| Author(s) | Year | Dataset title | Dataset URL | Database and Identifier |
|---|---|---|---|---|
| Garcia-Souto D, Diaz-Costas S, Bruzos AL, Rocha S, Pequeno A, Roman-Lewis CF, Costas D, Rodriguez-Castro J, Villanueva A, Silva L, Valencia JM, Annona G, Tarallo A, Richardo F, Bratos-Cetinic A, Posada D, Tubio JM | 2022 | Chamelea gallina mitochondrion, complete genome ECCG15_201 Healthy animal, foot | https://www.ncbi.nlm.nih.gov/nuccore/MW662591 | NCBI GenBank, MW662591 |
| Garcia-Souto D, Diaz-Costas S, Bruzo AL, Rocha AL, Pequeno A, Roman-Lewis CF, Costas D, Rodriguez-Castro J, Villanueva A, Silva L, Valencia JM, Annona G, Tarallo A, Richardo F, Bratos-Cetinic A, Posada D, Tubio JM | 2022 | Venus verrucosa mitochondrion, complete genome EMVV18_376 Normal mitochondria from neoplastic animal, foot | https://www.ncbi.nlm.nih.gov/nuccore/MW662592 | NCBI GenBank, MW662592 |
| Garcia-Souto D, Diaz-Costas S, Bruzos AL, Rocha S, Pequeno A, Roman-Lewis CF, Costas D, Rodriguez-Castro J, Villanueva A, Silva L, Valencia JM, Annona G, Tarallo A, Richardo F, Bratos-Cetinic A, Posada D, Tubio JM | 2022 | Venus verrucosa mitochondrion, complete genome EMVV18_391_Normal Normal mitochondria from neoplastic animal, foot | https://www.ncbi.nlm.nih.gov/nuccore/MW662595 | NCBI GenBank, MW662595 |
| Garcia-Souto D, Diaz-Costas S, Bruzo AL, Rocha A, Roman-Lewis CF, Costas D, Rodriguez-Castro J, Villanueva A, Silva L, Valencia JM, Annona G, Tarallo A, Richardo F, Bratos-Cetinic A, Posada D, Tubio JM, Pequeno A | 2022 | Venus verrucosa mitochondrion, complete genome EMVV18_395_Normal Normal mitochondria from neoplastic animal, foot | https://www.ncbi.nlm.nih.gov/nuccore/MW662597 | NCBI GenBank, MW662597 |
| Garcia-Souto D, Diaz-Costas S, Bruzos AL, Rocha S, Pequeno A, Roman-Lewis CF, Costas D, Rodriguez-Castro J, Villanueva A, Silva L, Valencia JM, Annona G, Tarallo A, Richardo F, Bratos-Cetinic A, Posada D, Tubio JM | 2022 | Venus verrucosa mitochondrion, complete genome EMVV18_400_Normal Normal mitochondria from neoplastic animal, foot | https://www.ncbi.nlm.nih.gov/nuccore/MW662599 | NCBI GenBank, MW662599 |

*Continued on next page*

*Continued*

| Author(s) | Year | Dataset title | Dataset URL | Database and Identifier |
|---|---|---|---|---|
| Garcia-Souto D, Diaz-Costas S, Bruzos AL, Rocha S, Pequeno A, Roman-Lewis CF, Costas D, Rodriguez-Castro J, Villanueva A, Silva L, Valencia JM, Annona G, Tarallo A, Richardo F, Bratoc-Cetinic A, Posada D, Tubio JM | 2022 | Venus verrucosa mitochondrion, complete genome ERVV17_2995_Normal Normal mitochondria from neoplastic animal, foot | https://www.ncbi.nlm.nih.gov/nuccore/MW662601 | NCBI GenBank, MW662601 |
| Garcia-Souto D, Diaz-Costas S, Bruzos AL, Rocha S, Pequeno A, Roman-Lewis CF, Costas D, Rodriguez-Castro J, Villanueva A, Silva L, Valencia JM, Annona G, Tarallo A, Richardo F, Bratos-Cetinic A, Posada D, Tubio JM | 2022 | Venus verrucosa mitochondrion, complete genome ERVV17_2997_Normal Normal mitochondria from neoplastic animal, foot | https://www.ncbi.nlm.nih.gov/nuccore/MW662602 | NCBI GenBank, MW662602 |
| Garcia-Souto D, Diaz-Costas S, Bruzos AL, Rocha S, Pequeno A, Roman-Lewis CF, Costas D, Rodriguez-Castro J, Villanueva A, Silva L, Valencia JM, Annona G, Tarallo A, Richardo F, Bratos-Cetinic A, Posada D, Tubio JM | 2022 | Venus verrucosa mitochondrion, complete genome ERVV17_3193_Normal Normal mitochondria from neoplastic animal, foot | https://www.ncbi.nlm.nih.gov/nuccore/MW662604 | NCBI GenBank, MW662604 |
| Garcia-Souto D, Diaz-Costas S, Bruzos AL, Rocha S, Pequeno A, Roman-Lewis CF, Costas D, Rodriguez-Castro J, Villanueva A, Silva L, Annona G, Tarallo A, Richardo F, Bratos-Cetinic A, Posada D, Tubio JM, Valencia JM | 2022 | Venus verrucosa mitochondrion, complete genome EVVV11_2_Normal Normal mitochondria from neoplastic animal, foot | https://www.ncbi.nlm.nih.gov/nuccore/MW662606 | NCBI GenBank, MW662606 |
| Garcia-Souto D, Diaz-Costas S, Bruzos AL, Rocha S, Pequeno A, Roman-Lewis CF, Costas D, Rodriguez-Castro J, Villanueva A, Silva L, Valencia JM, Annona G, Tarallo A, Richardo F, Bratos-Cetinic A, Posada D, Tubio JM | 2022 | Venus verrucosa mitochondrion, complete genome EMVV18_391_Tumour Transmissible neoplasia derived from Chamelea gallina, neoplastic hemocytes | https://www.ncbi.nlm.nih.gov/nuccore/MW662594 | GenBank, MW662594 |

*Continued on next page*

*Continued*

| Author(s) | Year | Dataset title | Dataset URL | Database and Identifier |
|---|---|---|---|---|
| Garcia-Souto D, Diaz-Costas S, Bruzos AL, Rocha S, Pequeno A, Roman-Lewis CF, Costas D, Rodriguez-Castro J, Villanueva A, Silva L, Valencia JM, Annona G, Tarallo A, Richardo F, Bratos-Cetinic A, Posada D, Tubio JM | 2022 | Venus verrucosa mitochondrion, complete genome EMVV18_395_ Tumour Transmissible neoplasia derived from Chamelea gallina, neoplastic hemocytes | https://www.ncbi.nlm.nih.gov/nuccore/MW662596 | NCBI GenBank, MW662596 |
| Garcia-Souto D, Diaz-Costas S, Bruzos AL, Rocha S, Pequeno A, Roman-Lewis CF, Costas D, Rodriguez-Castro J, Villanueva A, Silva L, Valencia JM, Annona G, Tarallo A, Richardo F, Bratos-Cetinic A, Posada D, Tubio JM | 2022 | Venus verrucosa mitochondrion, complete genome EMVV18_400_ Tumour Transmissible neoplasia derived from Chamelea gallina, neoplastic hemocytes | https://www.ncbi.nlm.nih.gov/nuccore/MW662598 | NCBI GenBank, MW662598 |
| Garcia-Souto D, Diaz-Costas S, Bruzos AL, Rocha S, Pequeno A, Roman-Lewis CF, Costas D, Rodriguez-Castro J, Villanueva A, Silva L, Valencia JM, Annona G, Tarallo A, Richardo F, Bratos-Cetinic A, Posada D, Tubio JM | 2022 | Venus verrucosa mitochondrion, complete genome ERVV17_2995_ Tumour Transmissible neoplasia derived from Chamelea gallina, neoplastic hemocytes | https://www.ncbi.nlm.nih.gov/nuccore/MW662600 | NCBI GenBank, MW662600 |
| Garcia-Souto D, Diaz-Costas S, Bruzos AL, Rocha S, Pequeno A, Roman-Lewis CF, Costas D, Rodrigues-Castro J, Villanueva A, Silva L, Valencia JM, Annona G, Tarallo A, Ricardo F, Bratos-Cetinic A, Posado D, Tubio JM | 2022 | Venus verrucosa mitochondrion, complete genome ERVV17_3193_ Tumour Transmissible neoplasia derived from Chamelea gallina, neoplastic hemocytes | https://www.ncbi.nlm.nih.gov/nuccore/MW662603 | NCBI GenBank, MW662603 |
| Garcia-Souto D, Diaz-Costa S, Bruzos AL, Rocha S, Pequeno A, Roman-Lewis CF, Costas D, Rodriguez-Castro J, Villanueva A, Silva L, Valencia JM, Annona G, Tarallo A, Richardo F, Bratos-Cetinic A, Posada D, Tubio JM | 2022 | Venus verrucosa mitochondrion, complete genome EVVV11_2_Tumour Transmissible neoplasia derived from Chamelea gallina, neoplastic hemocytes | https://www.ncbi.nlm.nih.gov/nuccore/MW662605 | NCBI GenBank, MW662605 |

*Continued on next page*

*Continued*

| Author(s) | Year | Dataset title | Dataset URL | Database and Identifier |
|---|---|---|---|---|
| Garcia-Souto D, Diaz-Costas S, Bruzo AL, Rocha S, Pequeno A, Roman-Lewis CF, Costas D, Rodriguez-Castro J, Villanueva A, Silva L, Valencia JM, Annona G, Tarallo A, Richardo F, Bratos-Cetinic A, Posada D, Tubio JM | 2022 | Chamelea gallina mitochondrion, complete genome IMCG15_69 Healthy animal, foot | https://www.ncbi.nlm.nih.gov/nuccore/MW662609 | NCBI GenBank, MW662609 |
| Garcia-Souto D, Diaz-Costas S, Bruzos AL, Rocha S, Pequeno A, Roman-Lewis CF, Costas D, Rodriguez-Castro J, Villanueva A, Silva L, Valencia JM, Annona G, Tarallo A, Richardo F, Bratos-Cetinic A, Posada D, Tubio JM | 2022 | Chamelea striatula mitochondrion, complete genome EVCS14_09 Healthy animal, foot | https://www.ncbi.nlm.nih.gov/nuccore/MW662611 | NCBI GenBank, MW662611 |

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
