## [Editor Report]

This paper describes a previously unknown lineage of transmissible cancer in a clam, in which the cancer arose from a different, but related, species. The data are clear and overall, the conclusions well-supported, and this finding increases our understanding of transmissible cancers in nature and will be of broad interest.

---

## [Decision Letter]

**Decision letter after peer review:**

Thank you for submitting your article "Mitogenome sequencing of marine leukemias reveals cancer contagion between clam species in the Seas of Southern Europe" for consideration by *eLife*. Your article has been reviewed by 3 peer reviewers, including Michael J Metzger as the Reviewing Editor and Reviewer #1, and the evaluation has been overseen by Detlef Weigel as the Senior Editor. The following individuals involved in review of your submission have agreed to reveal their identity: Ariberto Fassati (Reviewer #2); Nicolas Bierne (Reviewer #3).

Essential revisions:

There are a few sections where there are either unclear methods or the methods do not quite match the descriptions of the results.

1. Further explanation is required for the methods of the initial analysis of cox1 sequences.

2. There appears to be inconsistency between the descriptions of the de novo mitogenome assembly in the methods and in the results. Please clarify what methods were done and which samples were used for de novo mitogenome assemblies.

3. The statement that no nucleotide divergence was identified needs further explanation. It is unclear whether or how the authors looked for any additional somatic mutations within the cancer lineage in the DEAH12 locus, as the methods only describe analysis of SNVs that were fixed within each normal host population. The methods for mtDNA sequence extraction were also unclear and it is unclear whether the authors looked for somatic SNVs in the intergenic regions of the mtDNA in addition to the gene regions specified in the text. Since high-coverage whole genome sequence was generated, the claim can be read that no nucleotide divergence exists within the whole genomes-the statement should clarify which regions showed no evidence of nucleotide divergence and the methods should explain how the authors looked for SNVs.

4. The data for N1 animals requires further explanation. In particular, the data in the pie charts in Figure 2 do not allow the determination of whether the C. gallina cancer is completely absent or present with a low number of reads. It would help to know whether the number of C. gallina reads was greater than that found in analysis of normal samples and whether the tumor samples had more cancer-associated reads than the host samples. If the data are not consistent with a low level of C. gallina, then alternate explanations should be provided.

5. The authors present short-read sequencing at a considerable depth. Please explain further why only a single nuclear gene was chosen and what the criteria were for this selection.

6. Please include discussion of the divergence between V. verrucosa and C. gallina in the mitochondrial and nuclear regions analyzed. This would allow an appreciation of the genetic distance between the original host and the current host species. Additionally, if the high divergence prevented mapping all reads to a single mtDNA reference to analyze VAFs, please clarify this.

*Reviewer #1 (Recommendations for the authors):*

The manuscript is straightforward and overall the data support the conclusions. The two methods issues should be clarified and the claim about the lack of divergence between the cancer samples should be clarified as well.

*Reviewer #2 (Recommendations for the authors):*

The study would be strengthened by:

1. Analysing samples ERVV17-2997 and EMVV18-376 in greater depth.

2. Expanding the SNPs analysis to a few additional nuclear genes.

3. Citing correctly in the Introduction the first paper demonstrating the clonal origin of CTVT (Murgia et al. Cell 2006).

*Reviewer #3 (Recommendations for the authors):*

1. The mitogenome analysis would probably be more convincing by plotting variant allele fractions rather than mapping depth. Could you please explain if this was not possible, or why you prefered to present alternative mapping depth to the two reference mitogenomes? Alternatively provides VAF plots that, I think, will be more straightforward to catch for readers.

2. I really didn't understand how you came with analysing only a single nuclear gene. I initially thought at my first reading that you did genome skimming, but your sequencing coverage is very good. Out of the mountain came a mouse. You really need to explain what happened.

You said, “showed variant allele frequencies (VAF) at exclusively 0, 0.5 or 1.0 in all the sequenced healthy (non-neoplastic) specimens”. I didn’t understand what was the constrained imposed here, could you explain? You said, "A single gene was selected by these criteria?" which criteria exactly? How do you explain you selected only one gene given the data you have?

3. At the end you have two non-recombining loci, mitogenome and DEAH12. The multispecies coalescent approach was developed to analyze many independent genealogies, not just two. To my point of view this analysis bring nothing useful, and rather could be badly perceived by specialists.

4. In the abstract you said "leukemia-like transmissible cancers, called hemic neoplasia". I agree we have four decades of research on hemic/disseminated neoplasia, but only genetic analysis can prove it is a transmissible cancer (exactly what you did in this study) and it was never done before Metzger et al.'s Cell and Nature papers in 2015 and 2016. In addition, non-transmissible neoplasia have been described (in mussels: Yonetmitsu et al. 2019 *ELife*, Hammel et al. 2021 BioRXiv). Therefore, we don't know if studies that described hemic/disseminated neoplasia before dealt with a transmissible cancer, and we cannot equal hemic/disseminated neoplasia with transmissible cancer.

5. "beyond the death of the individual that spawned them" I would not use "spawned".

6. " hemic neoplasia has a clonal transmissible behaviour (Metzger et al., 2015), in which the haemocytes (i.e., the cells that populate the haemolymph and play a role in the immune response) are likely to be transmitted through marine water."

As said above, not all hemic neoplasia are transmissible. In addition, neoplastic cells are not necessarily haemocytes (probably not I'd say, but we don't know indeed).

7. "animals die because of the infection". Remissions have been described (see e.g. Burioli et al. JIP in mussels).

8. "leukemic haemocytes" -> leukemic cells.

9. “representing interesting models for the understanding of the genetic causes of cancer transmissibility and metastasis” Why? It adds a new twist on transmissibility but does not help to understand the first transmission. And metastasis? I don’t see why.

10. “Marine transmissible cancers likely move using ocean currents to colonize new regions”.

It's a little bit too imagery I'd say. Given how water is diffusive and how cancer cells sediment, it is much more likely that transmissible cancers spread progressively from host to nearby host, by small step.

11. "using functions in the R package" which functions? makefreq?

12. "Mitochondrial sequences for 13 coding genes and two rDNA genes from 16 specimens (six neoplastic, 10 non-neoplastic) were extracted from the paired-end sequencing data" Explanations are needed here. How did you sort the two mitogenomes in cancerous individuals? Was differential mapping sufficient? Or did you call variants? Or did you assemble the two mitogenomes with megahit?

13. It would be nice to provide a phylogentic tree with more venerid species from public databases, eg using COI-16S genes only. Could be a supplementary figure.

---

## [Author Response]

Essential revisions:There are a few sections where there are either unclear methods or the methods do not quite match the descriptions of the results.1. Further explanation is required for the methods of the initial analysis of cox1 sequences.

We retrieved a dataset of 3,745 sequences comprising all the barcode-identified venerid clam *Cox1* fragments available from the Barcode of Life Data System (BOLD, http://www.boldsystemns.org/). Redundancy was removed using CD-HIT (Fu, et al. 2012), applying a cut-off of 0.9 sequence identity, and sequences were trimmed to cover the same region. Whole-genome sequencing data from both healthy and tumoral warty venus clams was mapped onto this dataset, containing 118 venerid species-unique sequences, using BWA-mem, filtering out reads with mapping quality below 60 (-q60) and quantifying the overall coverage for each sequence with samtools idxstats. PCR primers were designed with Primer3 v2.3.7 (Koressaar, et al. 2018) to amplify a fragment of 354 bp from the *Cox1* mitochondrial gene of *V. verrucosa* and *C. gallina* (F: CCT ATA ATA ATT GGK GGA TTT GG, R: CCT ATA ATA ATT GGK GGA TTT GG). PCR products were purified with ExoSAP-IT and sequenced by Sanger sequencing.

We have included this new information in the methods section (page 22).

2. There appears to be inconsistency between the descriptions of the de novo mitogenome assembly in the methods and in the results. Please clarify what methods were done and which samples were used for de novo mitogenome assemblies.

In total, we performed whole-genome sequencing on 23 samples from 16 clam specimens, which includes eight neoplastic and eight non-neoplastic animals by Illumina paired-end libraries of 350 bp insert size and reads 150 bp long. First we assembled the mitochondrial genomes of one *V. verrucosa* (FGVV18_193), one *C. gallina* (ECCG15_201) and one *C. striatul*a (EVCS14_02) specimens with MITObim v1.9.1 (Hahn, et al. 2013), using gene baits from the following *Cox1* and *16S* reference genes to prime the assembly of clam mitochondrial genomes: V. verrucosa (*Cox1*, with GenBank accession number KC429139; and *16S*: C429301), *C. gallina* (*Cox1*: KY547757, *16S*: KY547777) and *C. striatula* (*Cox1*: KY547747, *16S*: KY547767). These draft sequences were polished twice with Pilon v1.23 (Walker, et al. 2014), and conflictive repetitive fragments from the mitochondrial control region were resolved using long read sequencing with Oxford Nanopore technologies (ONT) on a set of representative samples from each species and tumours. ONT reads were assembled with Miniasm v0.3 (Li 2016) and corrected using Racon v1.3.1 (Vaser, et al. 2017). Protein-coding genes, rDNAs and tRNAs were annotated on the curated mitochondrial genomes using MITOS2 web server (Bernt, et al. 2013), and manually curated to fit ORFs as predicted by ORF-FINDER (Rombel, et al. 2002). Then, we employed the entire mitochondrial DNAs of *V. verrucosa* (FGVV18_193) and *C. gallina* (ECCG15_201) as “references” to map reads from individuals with neoplasia, filter reads matching either mitogenome and assemble and polish their two (healthy and tumoral) mitogenomes individually as above. Further healthy individuals were later sequenced and their mitogenomes assembled, to further investigate the geographic and taxonomic spread of this neoplasia.

We have included this information in the methods section (page 21-22), and in the results (pages 7 and 8). mtDNA annotations are now shown in Figure 2—figure supplement 1. Nucleotide data for the mitochondrial DNA assemblies has been uploaded to GenBank under accession numbers MW662590-MW662611 and will be released upon publication or request.

3. The statement that no nucleotide divergence was identified needs further explanation. It is unclear whether or how the authors looked for any additional somatic mutations within the cancer lineage in the DEAH12 locus, as the methods only describe analysis of SNVs that were fixed within each normal host population. The methods for mtDNA sequence extraction were also unclear and it is unclear whether the authors looked for somatic SNVs in the intergenic regions of the mtDNA in addition to the gene regions specified in the text. Since high-coverage whole genome sequence was generated, the claim can be read that no nucleotide divergence exists within the whole genomes-the statement should clarify which regions showed no evidence of nucleotide divergence and the methods should explain how the authors looked for SNVs.

We obtained a preliminary nuclear assembly using short-reads only. Obviously, the resulting assemblies are fragmented and incomplete. This has limited the identification of candidate regions shared by the three genomes (*V. verrucosa* and both *Chamelea* clams). Out of the 44 candidate nuclear fragments we tested, only two (*DEAH12* and *TFHII*) turned out to give good PCR products, adequate for Sanger sequencing. As mentioned above, we now provide additional data on a second gene (*TFIIH*), identified and selected on the same basis as *DEAH12*. We find 14 and 15 sites, respectively, for the *DEAH12* and the *TFIIH* loci, with fixed SNVs (allele frequency >95%) that allowed to discriminate between the three relevant species (*V. verrucosa*, *C. gallina* and *C. striatula*) and the tumour. These diagnostic nucleotides were then used to filter the reads from individuals with neoplasia harbouring both DNA’s. Variation within the host lineage but not within the tumour was found along the nuclear DNA fragments employen in the ML phylogenies (see Author response image 1).

**Author response image 1. sa2fig1:** Molecular phylogenies based on the two selected nuclear markers. (a) DEAH12 gene and (b) TFIIH gene, and diagnostic loci discriminating among species and tumour. Bootstrap support values (500 replicates) from ML analyses above 50 are shown above the corresponding branches. Note all diagnostic nucleotides are identical between tumours (black dots).

Regarding the mtDNA, firstly, we assembled the mitochondrial genomes of one *V. verrucosa* (FGVV18_193), one *C. gallina* (ECCG15_201) and one *C. striatula* (EVCS14_02) specimens with MITObim v1.9.1 (Hahn, et al. 2013). Then, we employed the entire mitochondrial DNAs from *V. verrucosa* (FGVV18_193) and *C. gallina* (ECCG15_201) as “references” to map reads from individuals with neoplasia, filter reads matching either mitogenome and assemble and polish their two (healthy and tumoral) mitogenomes individually as above. Further healthy individuals were later sequenced and their mitogenomes assembled, to further investigate the geographic and taxonomic spread of this neoplasia. Despite the usefulness of the mitochondrial control region (CR) to detect differences among lineages, we refrained from using it for two reasons. (1) The CR shows considerable variation in both length and sequence among the three species, making their alignment difficult (in fact, previous phylogenetic studies based on whole mitochondrial DNA sequences in Veneridae excluded the CR: https://doi.org/10.1111/zsc.12454), and (2) the CR contains quasi-but-not-identical tandem repeats, as a other mollusks (i.e., the Venerid *Dosinia* clams https://doi.org/10.1371/journal.pone.0196466 or the *Littorina* marine snails https://doi.org/10.1016/j.margen.2016.10.006). In our case, repeats are larger than the short-reads insert size, and even though we could infer them by means of long read sequencing, polishing the resulting consensus sequences to overcome the intrinsic error rate of those lectures would yield inconclusive results, hindering the comparison between normal and tumoral haplotypes.

We updated the methods for the mitochondrial DNA analyses (pages 21-22) and the nuclear DNA analyses (page 23). We now include new data in the results and discussion (pages 9-10).

4. The data for N1 animals requires further explanation. In particular, the data in the pie charts in Figure 2 do not allow the determination of whether the C. gallina cancer is completely absent or present with a low number of reads. It would help to know whether the number of C. gallina reads was greater than that found in analysis of normal samples and whether the tumor samples had more cancer-associated reads than the host samples. If the data are not consistent with a low level of C. gallina, then alternate explanations should be provided.

Despite being sequenced at high coverage (30 Gb), we found no *C. gallina* reads in the sequenced N1 animals (ERVV17-2997 and ERVV18-373), most likely due to the presence of a low proportion of neoplastic cells in the haemolymph and the matched-normal tissue relative to normal cells. This is something common in some N1 tumours from hemic neoplasia, where only a few neoplastic cells are found.

We now provide more details on page 8.

5. The authors present short-read sequencing at a considerable depth. Please explain further why only a single nuclear gene was chosen and what the criteria were for this selection.

We obtained a preliminary nuclear assembly using short-reads only. Obviously, the resulting assemblies are fragmented and incomplete. This has limited the identification of candidate regions shared by the three genomes (*V. verrucosa* and both *Chamelea* clams). Out of the 44 candidate nuclear fragments we tested, only two (*DEAH12* and *TFHII*) turned out to give good PCR products, adequate for Sanger sequencing. As mentioned above, we now provide additional data on a second gene (*TFIIH*), identified and selected on the same basis as *DEAH12*. Individual ML phylogenies for these two fragments evidenced that tumours cluster together and separately from the host species and, in the case of DEAH12, closer to C. gallina. The MSC phylogeny was rebuilt including this new nuclear fragment.

In addition, we conducted a comparative screening of tandem repeats on the genomes of C. gallina and V. verrucosa. Two DNA satellites, namely CL4 and CL17, of, respectively, 332 and 429 bp monomer size, were very abundant in C. gallina and in the tumoral animals, but absent from all healthy V. verrucosa specimens. FISH probes designed for these satellites mapped on the heterochromatic regions, mainly in subcentromeric and subtelomeric positions, of both C. gallina and the neoplastic metaphases found in V. verrucosa, but were absent from the normal metaphases of the host species V. verrucosa. These results were consistent with the genomic abundance of these satellites in the NGS data and strongly suggest that these chromosomes derive from C. gallina.

We include the analysis of one additional nuclear locus, *TFIIH* (pages 9-10). We have obtained new ML and MSC phylogenies including this new locus (pages 9-10, figures 3b-c). Additional FISH approach looking for satellite DNA CL4 and CL11 was performed (page 10, figure 3d, Figure 3—figure supplement 1). The methods section has been updated accordingly (pages 20-21, 23-24).

6. Please include discussion of the divergence between V. verrucosa and C. gallina in the mitochondrial and nuclear regions analyzed. This would allow an appreciation of the genetic distance between the original host and the current host species. Additionally, if the high divergence prevented mapping all reads to a single mtDNA reference to analyze VAFs, please clarify this.

While the taxonomy and phylogeny of Venerids are under debate, *C. gallina* and *V. verrucosa* are well-differentiated species. Some authors consider that they belong to different subfamilies according to their external morphology, Chioninae and Venerinae (see World Register of Marine Species), while others consider that they pertain to *Venerinae* (Huber’s Compendium of Bivalves; Canappa et al. 1996; Chen et al. 2011). We have now estimated the genetic distance between these two species (23% in the assembled mtDNA fragment) and performed the following molecular phylogeny of 2045 venerids non-redundant COI gene sequences (see Author response image 2).

**Author response image 2. sa2fig2:** Molecular phylogenetic analysis by Maximum Likelihood method across Venerids. a) Maximum likelihood (ML) molecular phylogenetic tree of Veneridae based on a 323 bp alignment including 2045 non-redundant COI gene sequences recovered from Genbank/Boldsystems and all sequences derived from this work. Highlighted clades A and B according to Chen et al. 2011. Bootstrap support values (500 replicates) from ML analyses above 50 are shown above the corresponding branches. b) Zoom-in to the cluster of the venerid harbouring the neoplastic haplotype, both Chamelea species and Venus verrucosa in the previous tree (asterisk).

We have included new information on page 7.

Reviewer #2 (Recommendations for the authors):The study would be strengthened by:1. Analysing samples ERVV17-2997 and EMVV18-376 in greater depth.

Unfortunately, we were not able to perform further sequencing on these samples. We believe the absence of sequencing reads from the tumours is a consequence of a low proportion of neoplastic cells in the host tissues. Despite the high sequencing coverage obtained for the sequenced individuals, we did not find foreign reads in the N1 tumours (ERVV17-2997 and EMVV18-73) to mitochondrial nor nuclear (i.e., *DEAH12*, *TFHII*) level. This is most likely due to a very low proportion of neoplastic cells in their tissues.

We have added a sentence on page 8 that discuss this issue.

2. Expanding the SNPs analysis to a few additional nuclear genes.

We obtained a preliminary nuclear assembly using short-reads only. Obviously, the resulting assemblies are fragmented and incomplete. This has limited the identification of candidate regions shared by the three genomes (*V. errucose* and both *Chamelea* clams). Out of the 44 candidate nuclear fragments we tested, only two (*DEAH12* and *TFHII*) turned out to give good PCR products, adequate for Sanger sequencing. As mentioned above, we now provide additional data on a second gene (*TFIIH*), identified and selected on the same basis as *DEAH12*. Individual ML phylogenies for these two fragments evidenced that tumours cluster together and separately from the host species and, in the case of DEAH12, closer to C. gallina. The MSC phylogeny was rebuilt including this new nuclear fragment.

In addition, we conducted a comparative screening of tandem repeats on the genomes of C. gallina and V. errucose. Two DNA satellites, namely CL4 and CL17, of, respectively, 332 and 429 bp monomer size, were very abundant in C. gallina and in the tumoral animals, but absent from all healthy V. errucose specimens. FISH probes designed for these satellites mapped on the heterochromatic regions, mainly in subcentromeric and subtelomeric positions, of both C. gallina and the neoplastic metaphases found in V. errucose, but were absent from the normal metaphases of the host species V. errucose. These results were consistent with the genomic abundance of these satellites in the NGS data and strongly suggest that these chromosomes derive from C. gallina.

We include the analysis of one additional nuclear locus, *TFIIH* (pages 9-10). We have obtained new ML and MSC phylogenies including this new locus (pages 9-10, figures 3b-c). Additional FISH approach looking for satellite DNA CL4 and CL11 was performed (page 10, figure 3d, Figure 3—figure supplement 1). The methods section has been updated accordingly (pages 20-21, 23-24).

3. Citing correctly in the Introduction the first paper demonstrating the clonal origin of CTVT (Murgia et al. Cell 2006).

The reference is now included (page 4).

Reviewer #3 (Recommendations for the authors):1. The mitogenome analysis would probably be more convincing by plotting variant allele fractions rather than mapping depth. Could you please explain if this was not possible, or why you errucose to present alternative mapping depth to the two reference mitogenomes? Alternatively provides VAF plots that, I think, will be more straightforward to catch for readers.

Variant allele frequency (VAF) plots are a good approach to see the proportions of two cell types from the same species mapping onto the same genome. However, in this case, tumor and host cells do not map onto the same genome (mitochondrial genomes from these two species are far too divergent from each other, showing K2P nucleotide distance of 21.13%). In each tissue (haemolymph or foot) some read sequences are closer to the *C. gallina* (allegedly tumor cells) and others to *V. errucose* (ie. Host cells). Hence, VAF values will be mostly ~1 (haploid genome), not contributing to support or reject the hypothesis (see Author response image 3). However, mapping depth plots show how the proportion of reads of each tissue increase and decrease depending on the reference genome where the reads map (see Figure 2D).

**Author response image 3. sa2fig3:** For the warty venus neoplastic specimen EMVV18-400, the figure shows the comparison of the effects of plotting the variant allele frequency (VAF).

We included a sentence about the divergence between mtDNA genomes on page 7.

2. I really didn’t understand how you came with analysing only a single nuclear gene. I initially thought at my first reading that you did genome skimming, but your sequencing coverage is very good. Out of the mountain came a mouse. You really need to explain what happened.

We obtained a preliminary nuclear assembly using short-reads only. Obviously, the resulting assemblies are fragmented and incomplete. This has limited the identification of candidate regions shared by the three genomes (*V. errucose* and both *Chamelea* clams). Out of the 44 candidate nuclear fragments we tested, only two (*DEAH12* and *TFHII*) turned out to give good PCR products, adequate for Sanger sequencing. As mentioned above, we now provide additional data on a second gene (*TFIIH*), identified and selected on the same basis as *DEAH12*. Individual ML phylogenies for these two fragments evidenced that tumours cluster together and separately from the host species and, in the case of DEAH12, closer to C. gallina. The MSC phylogeny was rebuilt including this new nuclear fragment.

In addition, we conducted a comparative screening of tandem repeats on the genomes of C. gallina and V. errucose. Two DNA satellites, namely CL4 and CL17, of, respectively, 332 and 429 bp monomer size, were very abundant in C. gallina and in the tumoral animals, but absent from all healthy V. errucose specimens. FISH probes designed for these satellites mapped on the heterochromatic regions, mainly in subcentromeric and subtelomeric positions, of both C. gallina and the neoplastic metaphases found in V. errucose, but were absent from the normal metaphases of the host species V. errucose. These results were consistent with the genomic abundance of these satellites in the NGS data and strongly suggest that these chromosomes derive from C. gallina.

We include the analysis of one additional nuclear locus, *TFIIH* (pages 9-10). We have obtained new ML and MSC phylogenies including this new locus (pages 9-10, figures 3b-c). Additional FISH approach looking for satellite DNA CL4 and CL11 was performed (page 10, figure 3d, Figure 3—figure supplement 1). The methods section has been updated accordingly (pages 20-21, 23-24).

You said, “showed variant allele frequencies (VAF) at exclusively 0, 0.5 or 1.0 in all the sequenced healthy (non-neoplastic) specimens”. I didn’t understand what was the constrained imposed here, could you explain? You said, “A single gene was selected by these criteria?” which criteria exactly? How do you explain you selected only one gene given the data you have?

We now provide additional data on a second gene (*TFHII*), identified and selected on the same basis as *DEAH12*. Candidate genes were considered if they (1) were present in the genomes of the three species, and (2) showed variant allele frequencies (VAF) at exclusively 0, 0.5 or 1.0 in all the sequenced healthy (non-neoplastic) specimens. Regarding the 0, 0.5, 1.0 frequencies, we are looking for single-copy genes in the context of a diploid genome. Thus, for a given single-nucleotide variant (SNV) the allele frequency indicates the relative number of copies of one allele relative to the other. Homozygous SNVs have allele frequencies near 0 (AA) or 1 (BB). Heterozygous two-copy have allele frequencies near 0.5 (AB). Allelic imbalance results in intermediate values. For example, a SNV present in three copies will have four possible genotypes (AAA, AAB, ABB, BBB), thus giving possible allele frequencies of 0, 0.33, 0.67 or 1.

These criteria are defined in methods (page 23).

3. At the end you have two non-recombining loci, mitogenome and DEAH12. The multispecies coalescent approach was developed to analyze many independent genealogies, not just two. To my point of view this analysis bring nothing useful, and rather could be badly perceived by specialists.

As mentioned above (point 3), we now added additional data on another nuclear marker, as well as on two satellite DNAs. The multispecies coalescent approach can be applied to any number of loci, starting with one. A different thing is that the accuracy of the species tree estimate logically increases with the number of independent loci. Due to potential incomplete lineage sorting, we still prefer this kind of approach for two (now three) loci than a concatenated tree. Please, look at the single-locus analysis in Figure 3 and the preceding paragraph from the authors of BEAST at https://academic.oup.com/mbe/article/29/8/1969/1044583.

No specific action was taken for this query. However, we are happy to discuss this further if required by the reviewer.

4. In the abstract you said “leukemia-like transmissible cancers, called hemic neoplasia”. I agree we have four decades of research on hemic/disseminated neoplasia, but only genetic analysis can prove it is a transmissible cancer (exactly what you did in this study) and it was never done before Metzger et al.’s Cell and Nature papers in 2015 and 2016. In addition, non-transmissible neoplasia have been described (in mussels: Yonetmitsu et al. 2019 Elife, Hammel et al. 2021 BioRXiv). Therefore, we don’t know if studies that described hemic/disseminated neoplasia before dealt with a transmissible cancer, and we cannot equal hemic/disseminated neoplasia with transmissible cancer.

We agree with the reviewer that not all hemic neoplasia tumours are transmissible. We cannot change the abstract due to length restrictions, but we have changed the introduction to make this clear (“Some HNs have been proven to have a clonal transmissible behaviour…”).

We include a new sentence on page 4.

5. “beyond the death of the individual that spawned them” I would not use “spawned”.

Now the text reads as follows “…meaning that they can survive beyond the death of their hosts…” (page 4).

6. “ hemic neoplasia has a clonal transmissible behaviour (Metzger et al., 2015), in which the haemocytes (i.e., the cells that populate the haemolymph and play a role in the immune response) are likely to be transmitted through marine water.”As said above, not all hemic neoplasia are transmissible. In addition, neoplastic cells are not necessarily haemocytes (probably not I’d say, but we don’t know indeed).

Now the text reads as follows “…in which neoplastic cells, most likely haemocytes…” (page 4).

7. “animals die because of the infection”. Remissions have been described (see e.g. Burioli et al. JIP in mussels).

Now the text reads as follows “…although remissions have also been described (Burioli, et al. 2019)…” (page 4).

8. “leukemic haemocytes” -> leukemic cells.

Now the text reads as follows “…Despite the observation that leukemic cells are typically transmitted…” (page 4).

9. "representing interesting models for the understanding of the genetic causes of cancer transmissibility and metastasis" Why? It adds a new twist on transmissibility but does not help to understand the first transmission. And metastasis? I don't see why.

This has been removed from pages 4-5.

10. "Marine transmissible cancers likely move using ocean currents to colonize new regions".It's a little bit too imagery I'd say. Given how water is diffusive and how cancer cells sediment, it is much more likely that transmissible cancers spread progressively from host to nearby host, by small step.

This has been removed from pages 4-5.

11. "using functions in the R package" which functions? makefreq?

We screened for differentially fixed single-nucleotide variants between both species using the dapc function in the R package Exploratory Analysis of Genetic and Genomic Data adegenet.

A new sentence has been included on page 23.

12. "Mitochondrial sequences for 13 coding genes and two rDNA genes from 16 specimens (six neoplastic, 10 non-neoplastic) were extracted from the paired-end sequencing data" Explanations are needed here. How did you sort the two mitogenomes in cancerous individuals? Was differential mapping sufficient? Or did you call variants? Or did you assemble the two mitogenomes with megahit?

All healthy animals were assembled de novo using Mitobin and polished with Pilon. Two of them, *C. gallina* (ECCG15_201) and *V. verrucosa* (FGVV18_193), were initially obtained and used as representative sequences onto which short reads of the tumoral animals were mapped to. This differential mapping was enough to filter reads matching either mitogenome, and extract them from the original fastq files for all tumoral animals and assemble and polish neoplastic and normal mitogenomes individually for each specimen with neoplasia. Then, further healthy individuals (from *V. verrucosa*, *C. gallina* and *C. striatula*) were sequenced to investigate the evolutionary origins of this cancer.

All sequenced specimens/tissues are listed in Table 1. We also revised this in the results (pages 7-8) and methods (pages 21-22).

13. It would be nice to provide a phylogentic tree with more venerid species from public databases, eg using COI-16S genes only. Could be a supplementary figure.

While the taxonomy and phylogeny of Venerids are under debate, *C. gallina* and *V. verrucosa* are well-differentiated species. Some authors consider that they belong to different subfamilies according to their external morphology, Chioninae and Venerinae (see World Register of Marine Species), while others consider that they pertain to *Venerinae* (Huber’s Compendium of Bivalves; Canappa et al. 1996; Chen et al. 2011). We have now estimated the genetic distance among these two species (23% in the mtDNA). As for the phylogeny, we honestly believe this is a bit tricky to discuss in the main text, because it would break the flow of the story, and it would be out of the scope of this manuscript. However, we have performed the phylogeny, which is presented here for your review (see Author response image 2).

We have included new information on page 7.